# Using protection motivation theory to support patient adherence in healthcare settings: A scoping review

Olivia Hawksworth[1]*, Jemima Solt[2], Dowon Jang[2], Paul Norman[3], Daniel Hind[1], Raveen Jayasuriya[2]

1 Clinical Trials Research Unit, Sheffield Centre for Health and Related Research, School of Medicine and Population Health, University of Sheffield, United Kingdom, 2 School of Medicine and Population Health, University of Sheffield, Sheffield, United Kingdom, 3 Department of Psychology, University of Sheffield, Sheffield, United Kingdom

* o.hawksworth@sheffield.ac.uk

## Abstract

### Background

Protection motivation theory (PMT) shows promise as a basis for motivating healthy behaviours in healthcare settings. There has been no systematic overview of how PMT has been translated into clinical practice and which translation strategies effectively improve outcomes.

### Objectives

This scoping review aimed to systematically map and synthesise existing literature on PMT-based interventions targeting health behaviours in healthcare contexts.

### Methods

Medline, PsycINFO, and EMBASE were searched for studies applying PMT within healthcare contexts. Eligible populations had a clinical condition. To be eligible, studies had to report a healthcare-delivered PMT intervention with adherence outcomes directly benefiting participants. Two reviewers extracted data on study features, intervention characteristics including behaviour change techniques employed, PMT constructs addressed, results, and research recommendations. Findings were summarised narratively and tabulated.

### Results

Thirteen studies published between 1998 and 2023 met the eligibility criteria, including 12 randomised trials. Studies addressed acute and chronic conditions across primary, secondary, and non-clinical settings. Half significantly improved behaviour outcomes in intervention groups. All targeted coping-self-efficacy and perceived

**Data availability statement:** All relevant data are within the paper.

**Funding:** The author(s) received no specific funding for this work.

**Competing interests:** The authors have declared that no competing interests exist.

threat-severity PMT constructs to some degree. Combinations of behaviour change techniques did not clearly differentiate successful outcomes. Studies recommended longer follow-up, clarifying effective PMT component combinations, and drawing on multiple behaviour theories.

## Discussion

Despite heterogeneity in how PMT interventions were operationalised, they show potential benefits for motivating adherence to healthy behaviours. To enable optimisation and dissemination, consistent nonadherence detection and reporting methods are critical. Further research should include translation of PMT to other healthcare settings, refining methodology and implementing well-powered effectiveness trials in routine care.

---

## Introduction

Patient non-adherence to healthy behaviours and treatment plans is a widespread issue that constitutes both a humanistic and economic burden [1,2]. Studies show adherence rates can be as low as 30% in unsupervised home exercise programs and vary significantly across conditions [3]. This lack of adherence leads to worse health outcomes, increased risk of condition progression, greater healthcare use, and millions in avoidable costs [1,4,5]. Addressing this challenge requires us to understand the different reasons underlying non-adherence, including individual, socioeconomic, healthcare system, and treatment-related factors [5]. This, in turn, implies a need for theories, models and frameworks that can inform the development of interventions to improve adherence.

One such theory that has shown promise for improving adherence behaviours is Protection Motivation Theory (PMT) [6–8]. Meta-analyses have extensively validated PMT across various health contexts, demonstrating moderate to strong relationships between PMT variables and health behaviours (d+=0.52), with particularly strong effects for self-efficacy (d+=0.88) and notably higher effect sizes (d+=0.98) for medical treatment adherence specifically [8]. PMT considers how individuals react when encountering a health threat, with two key processes: threat appraisal and coping appraisal [7]. Threat appraisal refers to an individual's perception of their vulnerability to, and severity of, the health threat, as well as any intrinsic or extrinsic rewards associated with maladaptive behaviours. Coping appraisal refers to an individual's perception of their ability to carry out the recommended protective behaviour change (self-efficacy), the effectiveness of the behaviour to reduce the threat (response efficacy), and the response cost required to perform the behaviour. These two appraisal pathways contribute to an individual's protection motivation (i.e., intention) to perform the health behaviour (Fig 1). By targeting specific PMT constructs within an intervention, healthcare professionals and those designing interventions can aim to increase perceptions of threat and coping ability, motivating positive health behaviour changes.

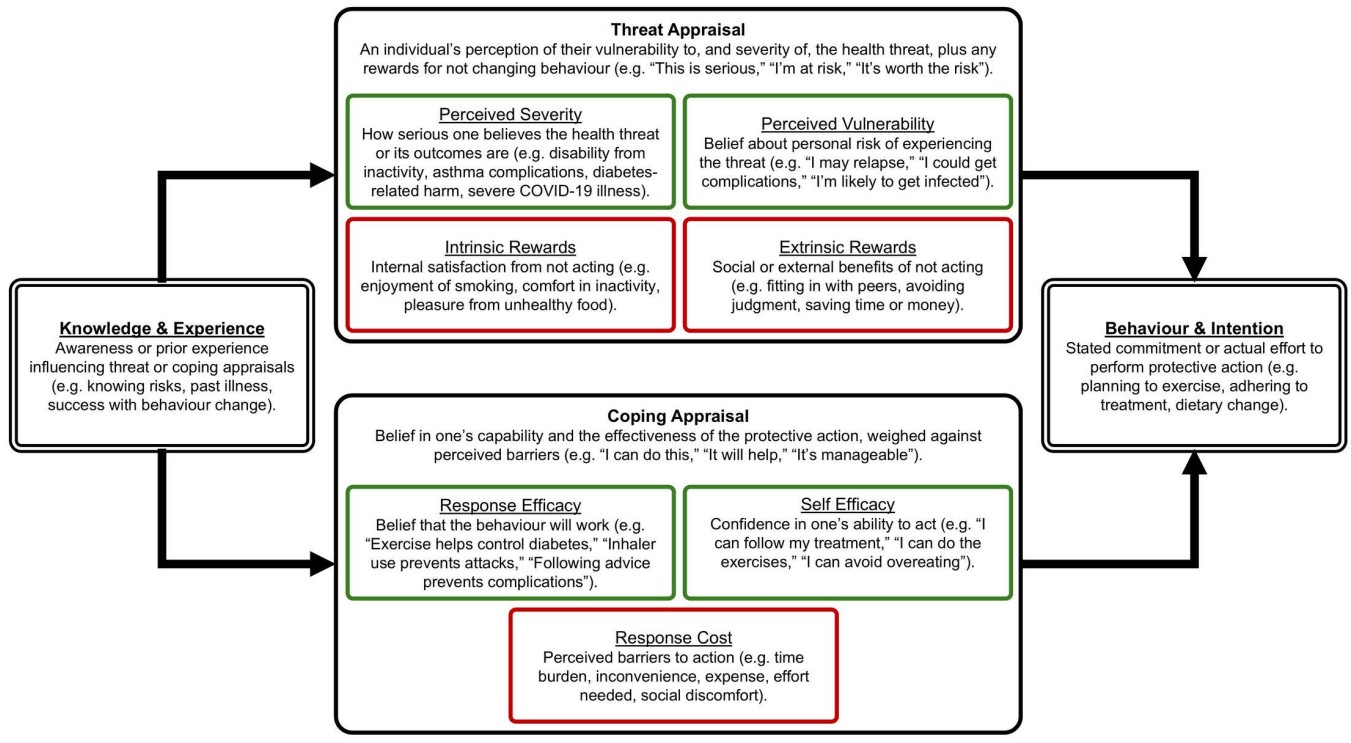

**Fig 1. Key constructs in Protection Motivation Theory.**

Guidelines on intervention design highlight the importance of reviewing existing evidence and theory throughout the development process [9]. Adaptation of existing interventions underpinned by robust theory represents an efficient approach, provided the mechanisms behind successful outcomes are retained [10]. While PMT's theoretical validity is well-established, uncertainty remains around how to effectively translate it into clinical practice. Reviewing existing PMT-based interventions can clarify which techniques effectively target relevant constructs to improve adherence, facilitating adaptation of successful interventions across healthcare domains.

A scoping review is an ideal vehicle to synthesise the existing literature on PMT interventions in healthcare. Scoping reviews can be used to map concepts and research activity across a broad area to clarify methods and identify gaps to shape future studies [11]. Our aim is not to assess effectiveness as in a systematic review, but to systematically characterise how PMT is being applied to improve adherence in physical healthcare contexts, providing practical guidance for front-line health professionals seeking to implement PMT-based interventions in routine care. Reviewing existing interventions based on PMT theory will reveal research gaps and concepts needing clarification to inform policy and the development of new optimised interventions tailored for wider implementation.

## Methods

### Protocol

We developed a protocol for the scoping review using the Preferred Reporting Items for Systematic Reviews and Meta-analysis Protocols (PRISMA-P) [12] which was fixed on 26th September 2023 and made available online [13].

## Eligibility criteria

The eligibility criteria for the scoping review were developed based on the Population Concept Context framework [11]. Eligible studies had clinical populations, including those at risk of condition progression. Studies of healthy populations were ineligible, including those who are currently healthy but at high risk of developing a condition (e.g., smokers, people with obesity). Studies in healthy pregnant people were ineligible. To be eligible in terms of concept, studies had to be of healthcare delivered interventions or adjunctive interventions based on PMT aiming to improve patient adherence. Protocol papers for studies of such interventions were included. Preventative interventions (e.g., for vaccine hesitancy, smoking cessation, breast screening behaviour) were ineligible. To be eligible, the context had to be such that the adherence outcome would directly affect the research participants. Studies in which the outcome of the intervention would not directly benefit the research participant (e.g., promoting adherence of condom use in HIV positive patients) were excluded.

## Information sources and search

On 18th September 2023 we searched MEDLINE, EMBASE (via Ovid) and PsycINFO using the search strategy detailed in Table 1. We limited the language to English but applied no date restrictions. In order to capture the grey literature, we conducted a search in Google Scholar. We screened the first 200 relevant references, as recommended in the literature [14].

## Selection of sources of evidence

We uploaded the search results to Rayyan [15] and removed duplicates. Two reviewers (JS and DJ) independently screened the titles and abstracts of all records against the eligibility criteria. The full texts of those records which appeared to be eligible were then assessed. Any disagreements on study selection were resolved by discussion with additional reviewers.

## Data charting process

Data charting forms were developed and piloted in Google Sheets. Two reviewers (JS and DJ) charted data from each eligible article. Any disagreements were resolved by discussion with additional reviewers.

## Data items

We charted data on the study characteristics (study design, setting, type of condition and type of outcomes collected) and the intervention characteristics. The data items relating to the intervention characteristics were informed by the Template for Intervention Description and Replication (TIDieR) checklist [16]. These were: the clinical outcomes ('why'); who provided the intervention; what procedures were employed (these were categorised using the cluster headings from the Behaviour Change Technique Taxonomy [17]); how the intervention was delivered; the number of times the intervention was delivered; tailoring and modification. We also charted data on which PMT constructs the interventions targeted, the results of the studies, and research recommendations.

**Table 1. Search strategy.**

|  | Search terms |
|---|---|
| 1. | Protection motivation theory.mp. |
| 2. | Exp Rehabilitation/ |
| 3. | Exp Surgery/ |
| 4. | 2 OR 3 |
| 5. | 1 AND 4 |

## Synthesis of results

We produced narrative and tabular summaries for study characteristics, intervention characteristics, PMT constructs targeted, results of the studies, and research recommendations.

## Results

### Selection of sources of evidence

After the removal of duplicates, 972 records were identified (Fig 2). Based on the title and abstract, 952 records were excluded and the full texts of the remaining 20 reports were assessed for eligibility. Seven of these were excluded. The remaining 13 studies were included in the review. No additional eligible studies were identified by screening the first 200 results from the Google Scholar search.

### Study characteristics

Studies included in the review were conducted in a range of countries: USA (n = 4 [18–21]); Iran (n = 3 [22–24]); China (n = 3 [25–27]); Canada (n = 1 [28]), New Zealand (n = 1 [29]), and the UK (n = 1 [30]). The types of studies were randomised controlled trials (RCTs) (n = 12 [18–27,29,30]) and a quasi-experimental study (n = 1 [28]). Studies took place in a range of settings: clinical primary care settings (n = 6 [19–23,29]); clinical secondary care settings (n = 4 [18,25,27,30]); and non-clinical settings (n = 3 [24,26,28]). The types of conditions being studied were chronic (n = 9 [18,19,21–23,25–27,30]) and acute (n = 4 [20,24,28,29]). The outcomes measured in the studies related to: beliefs, cognitions and knowledge

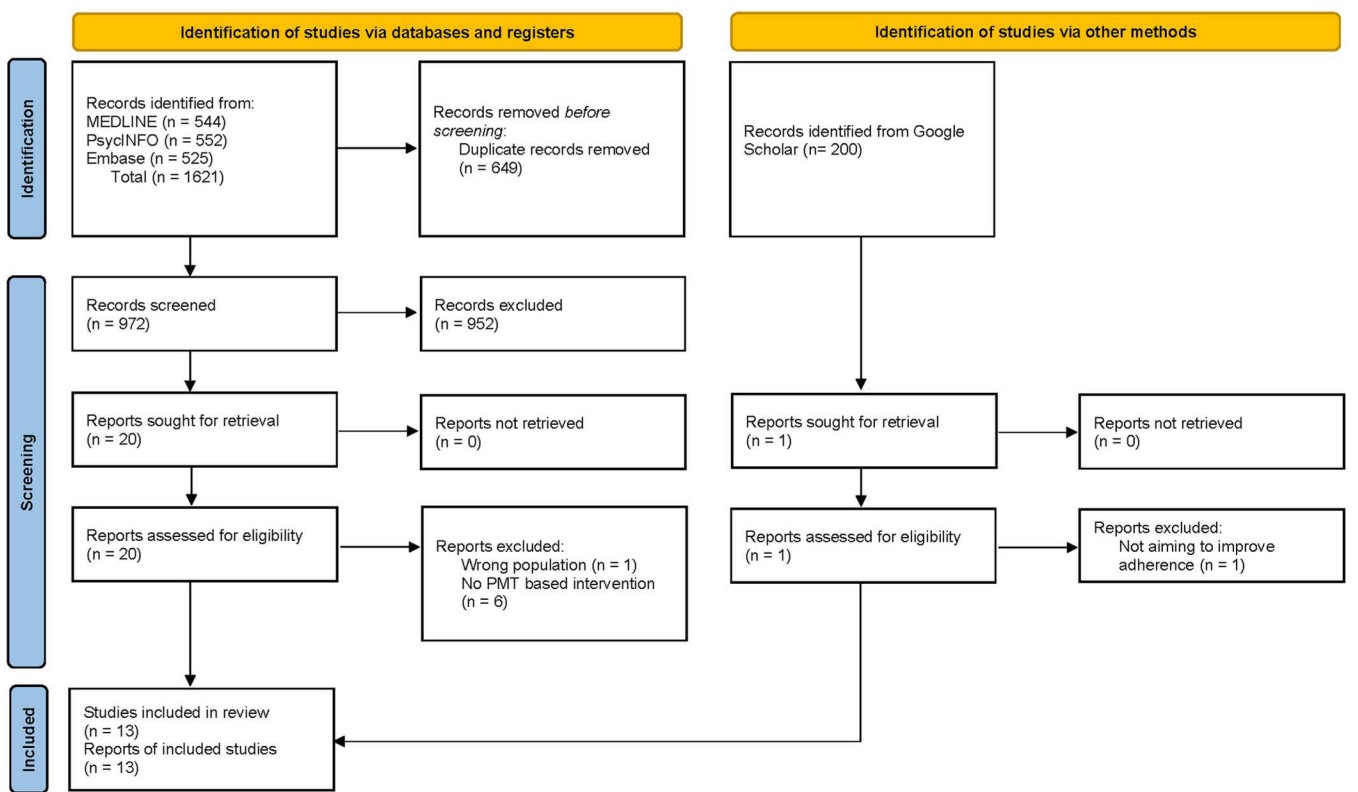

**Fig 2. Flow diagram of selection of sources of evidence for inclusion in the review.**

(n = 12 [18–22,24–30]); behaviour (n = 10 [18,20–26,28,29]); and physical health (n = 10 [18,20–23,25–29]). Table 2 presents the study characteristics.

## Characteristics of the interventions

The clinical aims of the interventions were to: improve self-management (n = 7 [18,20,21,25,27,29,30]); increase healthy behaviours (n = 4 [22–24,26]); and identify behaviour determinants (n = 2 [19,28]).

The materials used in the interventions were: digital educational materials (n = 3 [21,24,29]); paper-based educational materials (n = 6 [18–21,25,27]); educational materials (type unspecified) (n = 2 [22,23]); digital applications (n = 2 [26,30]); protocols, checklists, or lists to support intervention delivery (n = 3 [18,20,30]); and individualised risk information letters (n = 1 [28]).

We categorised the procedures used in each intervention according to the cluster headings listed in the Behaviour Change Technique Taxonomy [17]. These included: Natural consequences (n = 9 [18,19,22–27,29]); Feedback and monitoring (n = 5 [25–28,30]); Shaping knowledge (n = 5 [20–23,29]); Goals and planning (n = 3 [23,26,27]); Self-belief (n = 2

**Table 2. Study characteristics.**

| Author, year and country of origin | Design | Setting | Condition type | Outcomes | | |
|---|---|---|---|---|---|---|
| | | | | Beliefs/cognitions /knowledge | Behaviour | Physical health |
| Asimakopoulou, 2015 (UK) | RCT | Clinical, Secondary care | Chronic | PMT constructs; emotional reactions | None | None |
| Boeka, 2009 (USA) | RCT | Clinical, Secondary care | Chronic | PMT constructs; adherence intention | Adherence | Weight |
| Boeka, 2010 (USA) | RCT | Clinical, primary care | Chronic | PMT constructs; adherence intention | None | None |
| Dashti, 2020 (Iran) | RCT | Clinical, primary care | Chronic | PMT constructs | Dietary intake; physical activity | Weight; haemoglobin A1C |
| Guan, 2023 (China) | RCT | Clinical, Secondary care | Chronic | PMT Constructs | Chronic disease self-management | Pulmonary function |
| Haugtvedt, 1998 (USA) | RCT | Clinical, primary care | Acute | PMT constructs; adherence intention | Medication adherence | Illness symptoms |
| Lin, 2019 (China) | RCT (protocol) | Non-clinical | Chronic | PMT constructs; adherence intention | Adherence | Tuberculosis treatment outcomes; nicotine dependence |
| Morowatishari-fabad, 2021 (Iran) | RCT | Clinical, primary care | Chronic | None | Physical activity | V02 max; haemoglobin A1C |
| R Bassett, 2011 (Canada) | Quasi-experimental study | Non-clinical | Acute | PMT constructs | Physical activity | BMI; waist circumference; c-reactive protein; triglycerides; glucose; insulin |
| S Bassett, 2011 (New Zealand) | RCT | Clinical, primary care | Acute | PMT constructs; adherence intention | Rehabilitation adherence | Ankle function |
| Schaffer, 2004 (USA) | RCT | Clinical, primary care | Chronic | PMT constructs; knowledge | Medication adherence | Asthma control; quality of life |
| Vasli, 2023 (Iran) | RCT | Non-clinical | Acute | Knowledge; PMT constructs | Health behaviours | None |
| Yao, 2020 (China) | RCT | Clinical, Secondary care | Chronic | Self-rated depression; psychological resilience | None | Blood glucose control; quality of life |

RCT: randomised controlled trial

[19,23]); Associations (n = 1 [26]); Comparison of behaviour (n = 1 [29]); Regulation (n = 1 [21]); Repetition and substitution (n = 1 [26]) and Social support (n = 1 [25]). Characteristics of the interventions are summarised in Table 3.

The interventions were provided by: researchers (n = 8 [18,19,21,23,24,26,28,30]); allied health professionals (n = 2 [20,29]); multidisciplinary teams (n = 1 [25]), and nurses (n = 1 [27]). One study did not report who provided the intervention (n = 1 [22]).

Interventions were delivered in various ways: face-to-face only (n = 6 [18,19,22,23,27,30]); material led (n = 3 [21,28,29]); via digital applications (n = 2 [24,26]), and face-to-face and by telephone (n = 2 [20,25]). Interventions were

**Table 3. Characteristics of the interventions: brief name, why (clinical aim), what (materials), and what (procedures).**

| Study | Brief name | Why (clinical aim) | What (materials) | What (procedures) |
|---|---|---|---|---|
| Asimakopoulou 2015 | Individualised periodontal disease risk consultation to alter psychological variables related to adherence with periodontal instructions. | Improve self-management | Protocol/checklist/script to support intervention delivery; digital application | Feedback and monitoring |
| Boeka 2009 | PMT based informational intervention for people undergoing bariatric surgery to promote adherence to post-surgery eating behaviour guidelines | Improve self-management | Paper-based educational materials; protocol/checklist/script to support intervention delivery | Natural consequences |
| Boeka 2010 | PMT based psychosocial intervention for patients undergoing bariatric surgery. | Investigate behaviour determinants | Paper-based educational materials | Natural consequences; self-belief |
| Dashti 2020 | PMT based training programme for improving nutritional behaviour and physical activity in military patients with type 2 diabetes mellitus. | Promote healthy behaviours | Educational materials (unspecified) | Shaping knowledge; natural consequences |
| Guan 2023 | PMT based nursing intervention for patients with respiratory diseases. | Promote healthy behaviours | Paper-based educational materials | Natural consequences; feedback and monitoring; social support |
| Haugtvedt 1998 | PMT based brochure on taking antibiotics correctly, along with verbal reinforcement from a pharmacist, to promote adherence to an antibiotic regimen. | Improve self-management | Paper-based educational materials; protocol/checklist/script to support intervention delivery | Shaping knowledge |
| Lin 2019 | Smartphone application "QinTb" for smoking cessation in tuberculosis patients. | Promote healthy behaviours | Digital application | Natural consequences; goals and planning; feedback and monitoring; repetition and substitution; associations |
| Morowatishari-fabad 2021 | Educational intervention based on PMT and Implementation Intention to promote physical activity in patients with type 2 diabetes mellitus. | Promote healthy behaviours | Educational materials (unspecified) | Natural consequences; goals and planning; self-belief; shaping knowledge |
| R Bassett 2011 | Individualised, PMT based informational intervention to promote physical activity among people with spinal cord injury. | Investigate behaviour determinants | Individualised risk information letters | Feedback and monitoring |
| S Bassett 2011 | PMT based informational video to promote adherence to physiotherapy. | Improve self-management | Digital educational materials | Shaping knowledge; natural consequences; comparison of behaviour |
| Schaffer 2004 | A PMT based informational intervention to promote asthma medication adherence. | Improve self-management | Digital educational materials; paper-based educational materials | Shaping knowledge; regulation |
| Vasli 2023 | PMT based empowerment intervention to promote health behaviours in women with HPV. | Promote healthy behaviours | Digital educational materials | Natural consequences |
| Yao 2020 | PMT based informational intervention for patients with type 2 diabetes mellitus. | Improve self-management | Paper-based educational materials | Feedback and monitoring; natural consequences; goals and planning |

delivered: individually (n = 6 [20,24–26,28,30]); in both groups and individually (n = 4 [18,19,27,29]) and in groups (n = 1 [23]). Two studies did not report whether the intervention was delivered individually or in groups (n = 2 [21,22]).

Interventions were delivered in: a single session (n = 6 [19–21,28–30]); more than five sessions (n = 4 [23,25–27]) and fewer than five sessions (n = 3 [18,22,24]). These characteristics of the intervention are summarised in Table 4.

As detailed in Table 5, six of the interventions in the studies were tailored to individual participants (n = 6 [22,23,25–27,29]) and seven were not (n = 7 [18–21,24,28,30]).

Most studies did not report any modifications to the intervention (n = 12 [18–28,30]). One study increased the emphasis on self-efficacy within the intervention after seeing no significant difference between the intervention group and the control group [29].

Few studies reported on approaches to maintain intervention fidelity – i.e., how well (planned) (n = 3 [18,20,25]). The approaches that were reported are summarised in Table 5. None of the studies reported on the actual intervention fidelity – i.e., how well (actual).

### Targeting of PMT constructs

The studies targeted different constructs within PMT (Table 6). Interventions targeted the following threat appraisal constructs: perceived vulnerability (n = 10 [18,19,21,23–26,28–30]); perceived severity (n = 10 [18,19,22–27,29,30]); and intrinsic and extrinsic rewards (n = 4 [20,23–25]). One study did not report targeting any threat appraisal constructs (n = 1 [20]). In terms of coping appraisal, interventions targeted: self-efficacy (n = 10 [18–21,23–27,30]); response efficacy (n = 8 [18,20–22,24,25,27,29]); response cost (n = 4 [23–26]). One study did not report targeting any coping appraisal constructs (n = 1 [28]).

### Follow up

The length of follow up ranged from immediately post intervention (n = 2, [27,30]) to 12 months post intervention (n = 1, [26]). Table 7 presents the length of follow up used in each of the studies.

**Table 4. Characteristics of the interventions: who provided, how, and number of sessions.**

| Study | Who provided | How | | Number of sessions |
|---|---|---|---|---|
| | | Mode of delivery | Group or individual delivery | |
| Asimakopoulou 2015 | Researchers | Face-to-face | Individual | 1 |
| Boeka 2009 | Researchers | Face-to-face | Individual and group | <5 |
| Boeka 2010 | Researchers | Face-to-face | Individual and group | 1 |
| Dashti 2020 | Unspecified | Face-to-face | Not reported | <5 |
| Guan 2023 | Multidisciplinary team | Face-to-face; telephone | Individual | >5 |
| Haugtvedt 1998 | Pharmacists | Face-to-face; telephone | Individual | 1 |
| Lin 2019 | Researchers | Virtual | Individual | >5 |
| Morowatisharifabad 2021 | Researchers | Face-to-face | Group | >5 |
| R Bassett 2011 | Researchers | Material led | Individual | 1 |
| S Bassett 2011 | Physiotherapists | Material led | Individual and group | 1 |
| Schaffer 2004 | Researchers | Material led | Not reported | 1 |
| Vasli 2023 | Researchers | Virtual | Individual | <5 |
| Yao 2020 | Nurses | Face-to-face | Individual and group | >5 |

**Table 5. Characteristics of the interventions: tailoring, modification, how well (planned) and how well (actual).**

| Study | Tailoring | Modification | How well | |
|---|---|---|---|---|
| | | | Planned | Actual |
| Asimakopoulou 2015 | No | None reported | Not reported | Not reported |
| Boeka 2009 | No | None reported | Investigator used a checklist of discussion points to maintain continuity across the group discussions | Not reported |
| Boeka 2010 | No | None reported | Not reported | Not reported |
| Dashti 2020 | Yes | None reported | Not reported | Not reported |
| Guan 2023 | Yes | None reported | The intervention team were trained in PMT before delivering the intervention | Not reported |
| Haugtvedt 1998 | No | None reported | Pharmacists followed a written protocol for verbal reinforcement | Not reported |
| Lin 2019 | Yes | None reported | Not reported | Not applicable – study protocol |
| Morowatishari-fabad 2021 | Yes | None reported | Not reported | Not reported |
| R Bassett 2011 | No | None reported | Not reported | Not reported |
| S Bassett 2011 | Yes | Increased emphasis on self-efficacy within the intervention. | Not reported | Not reported |
| Schaffer 2004 | No | None reported | Not reported | Not reported |
| Vasli 2023 | No | None reported | Not reported | Not reported |
| Yao 2020 | Yes | None reported | Not reported | Not reported |

## Results of the interventions

Seven studies found a significant difference in behaviour change between the intervention and control group [18,21,22,24,25,27,30]. Five studies found no significant difference in behaviour change between the intervention and control group [19,20,23,28,29]. One of the included articles was a protocol for a study so did not present any results [26].

Studies that found a significant difference in behaviour change between the intervention and control group more often targeted the response efficacy construct (7/7 [18,21,22,24,25,27,30]) than those which did not find significant differences (2/5 [20,29]).

## Research recommendations

Studies provided a range of recommendations for further research. These were: using longer follow up periods (n = 6 [18,23,25,27,28,30]); investigating the impact of specific PMT constructs (n = 3 [19,20,28]); incorporating other behaviour change theories alongside PMT (n = 2 [21,29]); investigating the impact of delivering the intervention over a longer period of time (n = 2 [27,30]); exploring the optimal mode of intervention delivery (n = 1 [29]) and using more individualised interventions (n = 1 [28]). These are summarised in Table 8.

## Discussion

This scoping review of PMT-based healthcare interventions included 13 studies published between 1998 and 2023 in six different countries. The majority were randomised controlled trials conducted across primary care, secondary care, and non-clinical settings for both acute and chronic conditions. Intervention aims included promoting healthy behaviours, improving self-management, and identifying behaviour determinants. Half of studies reported significant between-group differences in behaviour change, while the remainder found no significant differences. The core PMT constructs

**Table 6. Threat appraisal and coping appraisal constructs targeted in each intervention.**

| Study | Threat appraisal | | | Coping appraisal | | |
|---|---|---|---|---|---|---|
| | Perceived severity | Perceived vulnerability | Intrinsic and extrinsic rewards | Response efficacy | Self-efficacy | Response cost |
| Asimakopoulou 2015 | ✓ | ✓ | | ✓ | ✓ | |
| Boeka 2009 | ✓ | ✓ | | ✓ | ✓ | |
| Boeka 2010 | ✓ | ✓ | | | ✓ | |
| Dashti 2020 | ✓ | | | ✓ | | |
| Guan 2023 | ✓ | ✓ | ✓ | ✓ | ✓ | ✓ |
| Haugtvedt 1998 | | | ✓ | ✓ | ✓ | |
| Lin 2019 | ✓ | ✓ | | | ✓ | |
| Morowatisharifabad 2021 | ✓ | ✓ | ✓ | | ✓ | ✓ |
| R Bassett 2011 | | ✓ | | | | |
| S Bassett 2011 | ✓ | ✓ | | ✓ | | |
| Schaffer 2004 | ✓ | ✓ | | ✓ | ✓ | |
| Vasli 2023 | ✓ | ✓ | ✓ | ✓ | ✓ | ✓ |
| Yao 2020 | ✓ | | | ✓ | ✓ | |

**Table 7. Length of follow up used in each of the studies.**

| Study | Length of follow up |
|---|---|
| Asimakopoulou 2015 | Immediately post intervention |
| Boeka 2009 | 3 months |
| Boeka 2010 | 1 week |
| Dashti 2020 | 3 months |
| Guan 2023 | 4 weeks |
| Haugtvedt 1998 | 10 days |
| Lin 2019 | 12 months |
| Morowatisharifabad 2021 | 3 months |
| R Bassett 2011 | 2 weeks |
| S Bassett 2011 | Not reported |
| Schaffer 2004 | 6 months |
| Vasli 2023 | 3 months |
| Yao 2020 | Immediately post intervention |

of perceived severity, perceived vulnerability, response efficacy and self-efficacy were targeted in most interventions, whereas response costs and intrinsic and extrinsic rewards were targeted in few interventions. Few studies reported on intervention fidelity. We could not identify any consistent intervention feature clearly associated with superior outcomes. While showing potential promise in select settings, this review revealed substantial heterogeneity in how PMT has been implemented and substantial remaining uncertainties around what drives efficacy to motivate patient adherence.

Strengths of this scoping review include the use of robust methodology aligned with published guidance. The involvement of two independent reviewers improved reliability in study selection and data extraction. The TIDieR framework enabled consistent reporting on key features [16]. The broad, cross-disciplinary scope facilitates identification of research gaps and potential new applications.

However, as a scoping review, there was no formal quality assessment or risk of bias evaluation as in a systematic review [11]. This is especially important in interpreting the results, given the small size of the individual studies, which is

**Table 8. Research gaps identified from included studies.**

| Theme | Research Gap | Importance | References |
|---|---|---|---|
| Longer follow-up periods | Are behaviour changes following the intervention maintained? | Examining the long-term effects is important for healthcare planning and for understanding the effectiveness of the intervention. | [18,23,25,27,28,30] |
| Specific constructs of PMT | Does emphasis on specific combinations of PMT constructs impact the success of an intervention? | To refine the design of interventions to produce optimal behaviour change. | [19,20,28] |
| Other behaviour change theories | What is the effect of incorporating other behaviour change theories with PMT? | A multi-theoretical approach may create more effective interventions for patient adherence. | [21,29] |
| Mode of intervention delivery | What other modes of presenting the intervention may improve behaviour change? | To increase effectiveness and improve efficiency of interventions. | [29] |
| Individualising interventions | Does tailoring of an intervention to each individual produce greater or faster behaviour change? | To understand how to make the most effective intervention. | [28] |
| Longer duration of intervention delivery | Does increased duration of intervention delivery affect behaviour change? | To optimise intervention design and delivery. | [27,30] |

often associated with exaggerated treatment effects [31–35]. The included studies exhibited substantial heterogeneity in populations, interventions, outcomes, and PMT operationalisation, precluding definitive conclusions or clinical recommendations on what drives efficacy. Restricting the search to three databases may have excluded potentially relevant literature. Generalisability is limited given that all interventions were trialled in upper-middle- or high-income countries. Finally, the lack of primary studies investigating adaptations of successful PMT interventions represents a broader evidence gap.

The synthesis provides a breakdown of the existing PMT based interventions in a range of clinical areas, allowing healthcare professionals to select components that are relevant to and practicable in their setting. Those developing PMT interventions should follow best practice guidance on design [9,36] and reporting [16,37]. Healthcare professionals should consider assessing perceptions of threat and coping ability using simple PMT screening questions to identify targets for intervention. The most relevant constructs within PMT are likely to differ across health conditions and treatment regimens. Therefore, exploratory research should be undertaken prior to intervention development to identify the constructs most pertinent to the specific clinical context. Multi-component interventions addressing various PMT constructs may have the greatest impact on behavioural intentions and health outcomes. Adaptations maximising perceived personal relevance are likely to be important for success. Cultural factors should be taken into account when designing and adapting interventions to ensure that these are relevant for the target audience.

This scoping review demonstrates that PMT holds promise for motivating health behaviours, but uncertainty remains around optimal implementation. Most studies revealed positive effects in select settings, but none established definitive strategies for ensuring reliable adherence improvements. Future research should investigate effective combinations of constructs, delivery adaptations and individualisation approaches to expand impact and accessibility. Studies should determine necessary intervention intensity and duration, and examine intervention sustainability in well-powered, real-world effectiveness trials. Above all researchers should use standardised reporting methods and pragmatic, theory-driven trials assessing long-term, patient important outcomes across diverse contexts.

## Supporting information

**S1 File. PRISMA-ScR Checklist.**
(PDF)

## Author contributions

**Conceptualization:** Paul Norman, Daniel Hind, Raveen Jayasuriya.

**Investigation:** Olivia Hawksworth, Jemima Solt, Dowon Jang.

**Supervision:** Paul Norman, Daniel Hind, Raveen Jayasuriya.

**Writing – original draft:** Olivia Hawksworth.

**Writing – review & editing:** Jemima Solt, Dowon Jang, Paul Norman, Daniel Hind, Raveen Jayasuriya.

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
