## [Decision Letter · Decision Letter 0]

1 Dec 2024

PONE-D-24-22494Using Protection Motivation Theory to support patient adherence in healthcare settings: A scoping reviewPLOS ONE

Dear Dr. Hawksworth,

Thank you for submitting your manuscript to PLOS ONE. After careful consideration, we feel that it has merit but does not fully meet PLOS ONE’s publication criteria as it currently stands. Therefore, we invite you to submit a revised version of the manuscript that addresses the points raised during the review process.

Please work on a major revision of the manuscript based on reviewers' feedback. As both reviewers have pointed out, the significance of the study needs to be further strengthened. Significant extensions of the methodological framework seem necessary based on both reviewers' comments, which the editor also tends to agree with after reviewing the paper.

We look forward to receiving your revised manuscript.

Kind regards,

Chenfeng Xiong

Academic Editor

PLOS ONE

2. As required by our policy on Data Availability, please ensure your manuscript or supplementary information includes the following:

Reviewers' comments:

Reviewer's Responses to Questions

**Comments to the Author**

1. Is the manuscript technically sound, and do the data support the conclusions?

Reviewer #1: Yes

Reviewer #2: Partly

2. Has the statistical analysis been performed appropriately and rigorously? 

Reviewer #1: Yes

Reviewer #2: I Don't Know

3. Have the authors made all data underlying the findings in their manuscript fully available?

Reviewer #1: Yes

Reviewer #2: Yes

4. Is the manuscript presented in an intelligible fashion and written in standard English?

Reviewer #1: Yes

Reviewer #2: Yes

5. Review Comments to the Author

Reviewer #1: Thank you very much for the opportunity to review this article. It provides many interesting insights, and I have a few small questions:

1. The introduction needs to better highlight the significance of the study; it seems insufficiently expressed in this regard.

2. There are several technical terms in the article that appear to be self-explanatory, such as "threat appraisal" and "coping appraisal." I believe the authors should provide more explanation for these terms.

3. The article mainly focuses on a simple description of previous studies. Although I was looking forward to seeing a classification of the PMT theoretical framework or some other kind of organization, it seems this is missing.

4. Additionally, some previous research has highlighted the impact of cultural context on PMT. Given that the sample size in this study is not large, I think the article could benefit from further exploration and extension from a cultural perspective.

Reviewer #2: 1. The significance of the research is not prominent enough. Although the research topic is interesting, it needs to be emphasized why this topic is important and worth studying.

2. While the eligibility criteria are detailed and the sample inclusion appears well-defined, I wonder if 13 studies are sufficient to support a comprehensive review. For researchers familiar with Protection Motivation Theory (PMT) in the context of patients and health adherence, many would likely already be aware of these studies.

3. Consider a Meta-Analysis on PMT: Would a meta-analysis on PMT offer greater value? For example, exploring whether conflicting relationships exist between perceived threat, perceived efficacy, and health behaviors could yield deeper insights.

4. Your appendix currently lists data collection and exclusion criteria in a basic format. It would be more informative to include the specific standards directly in this figure.

6. PLOS authors have the option to publish the peer review history of their article (what does this mean? ). If published, this will include your full peer review and any attached files.

**Do you want your identity to be public for this peer review?** For information about this choice, including consent withdrawal, please see our Privacy Policy .

Reviewer #1: No

Reviewer #2: No

---

## [Author Response · Author response to Decision Letter 1]

31 Jan 2025

This information has also been provided as a separate file with the revised manuscript.

Dear Editor,

Thank you for the opportunity to submit a revised draft of the manuscript “Using Protection Motivation Theory to support patient adherence in healthcare settings: A scoping review” for publication in PLOS ONE. We have incorporated most of the suggestions made by the reviewers and these are highlighted within the manuscript. Please see below a point-by-point response to the reviewers’ comments. All line numbers refer to the Revised Manuscript with Track Changes file.

Reviewer #1: Thank you very much for the opportunity to review this article. It provides many interesting insights, and I have a few small questions:

1. The introduction needs to better highlight the significance of the study; it seems insufficiently expressed in this regard.

Thank you for highlighting this. We have amended the Introduction to better express the significance and aim of the study:

“Studies show adherence rates can be as low as 30% in unsupervised home exercise programs and vary significantly across conditions (Ley and Putz 2024).” (Lines 45-46)

“Meta-analyses have extensively validated PMT across various health contexts, demonstrating moderate to strong relationships between PMT variables and health behaviours (d+ = 0.52), with particularly strong effects for self-efficacy (d+ = 0.88) and notably higher effect sizes (d+ = 0.98) for medical treatment adherence specifically (Floyd et al., 2000).” (Lines 53-56)

“While PMT's theoretical validity is well-established, uncertainty remains around how to effectively translate it into clinical practice.” (Lines 71-72)

“Our objective is not to re-validate PMT but rather to systematically map how PMT is being applied to improve adherence in physical healthcare contexts, providing practical guidance for frontline health professionals seeking to implement PMT-based interventions in routine care.” (Lines 77-81)

2. There are several technical terms in the article that appear to be self-explanatory, such as "threat appraisal" and "coping appraisal." I believe the authors should provide more explanation for these terms.

We agree with the reviewer’s comment and have added some further explanation of these terms in the Introduction (Lines 58-62):

“Threat appraisal refers to an individual’s perception of their vulnerability to, and severity of, the health threat, as well as any intrinsic or extrinsic rewards associated with maladaptive behaviours. Coping appraisal refers to an individual’s perception of their ability to carry out the recommended protective behaviour change (self-efficacy), the effectiveness of the behaviour to reduce the threat (response efficacy), and the response cost required to perform the behaviour.”

3. The article mainly focuses on a simple description of previous studies. Although I was looking forward to seeing a classification of the PMT theoretical framework or some other kind of organization, it seems this is missing.

In the scoping review we have focused on interventions that have drawn on PMT to increase patient adherence in healthcare settings. PMT therefore provides the theoretical framework for the review. We were unclear whether the reviewer also wanted us also to classify the PMT (and in what way). PMT is a model of the social cognitive determinants of health behaviour (Conner & Norman, 2015) that, for example, predominantly focuses on the motivation component of the COM-B model of behaviour change (Michie et al., 2011). We have not included this information in the revised submission, but would be happy to do so if the reviewer feels that it is important to do so.

4. Additionally, some previous research has highlighted the impact of cultural context on PMT. Given that the sample size in this study is not large, I think the article could benefit from further exploration and extension from a cultural perspective.

We agree that the impact of cultural context on PMT is an important consideration. We feel that an in-depth exploration of this goes beyond the scope of this review, however, we have added some additional details on this in the manuscript.

Firstly, we have added the country of origin to the Results (Lines 141-2) and in Table 1:

“Studies included in the review were conducted in a range of countries: USA (n=4); Iran (n=3); China (n=3); Canada (n=1), New Zealand (n=1), and the UK (n=1).”

Secondly, the Discussion identifies a lack of generalisability to lower income countries as a limitation of the review (Lines 251-2).

“Generalisability is limited given that all interventions were trialled in upper-middle- or high-income countries”.

Thirdly, we have commented on the consideration of cultural context in the Discussion (Lines 260-1):

“Cultural factors should be taken into account when designing and adapting interventions to ensure that these are relevant for the target audience.”.

Reviewer #2:

1. The significance of the research is not prominent enough. Although the research topic is interesting, it needs to be emphasized why this topic is important and worth studying.

Thank you for this comment. The need to emphasise the significance of the research was also raised by Reviewer 1. We believe we have addressed this comment in our response to Reviewer 1’s first comment.

2. While the eligibility criteria are detailed and the sample inclusion appears well-defined, I wonder if 13 studies are sufficient to support a comprehensive review. For researchers familiar with Protection Motivation Theory (PMT) in the context of patients and health adherence, many would likely already be aware of these studies.

We believe that the review makes a valuable contribution to knowledge in this area, even with only 13 studies. The review maps the existing evidence on the use of PMT-based interventions to improve patient adherence in healthcare settings and its results provide context-specific, actionable knowledge about the application of PMT.

The study is aimed at healthcare professionals who are interested in improving patient adherence in order to improve clinical outcomes. As such, the target audience would likely not be aware of these studies. In addition, synthesising the literature in this way adds value even for those who are aware of some of the individual studies as it provides an overview of the range of ways that PMT has been applied in this context. The target audience of the review has been clarified in the manuscript (Lines 79-80):

“…providing practical guidance for frontline health professionals seeking to implement PMT-based interventions in routine care.”

3. Consider a Meta-Analysis on PMT: Would a meta-analysis on PMT offer greater value? For example, exploring whether conflicting relationships exist between perceived threat, perceived efficacy, and health behaviors could yield deeper insights.

Existing meta-analyses have validated PMT across various health contexts. We sought not to re-validate PMT, but rather to map the ways that it has been applied to improve adherence in physical healthcare contexts. We opted to conduct a scoping review as these are designed to map the available literature on a topic, identify knowledge gaps, and summarise key findings across a variety of study types. They do not provide a quantitative synthesis of the data as a meta-analysis would. We hope that the amended introduction better explains this decision.

We agree that it would be valuable to understand whether any particular characteristics of the interventions were associated with those studies which found significant differences between intervention and control groups. The main difference between the studies that found significant vs non-significant differences appears to relate to whether they targeted the response efficacy construct. All of those studies which found significant differences targeted response efficacy, compared to 2/5 of those which did not find significant differences.

We have added the following text to the Results (Lines 213-5) to reflect this:

“Studies that found a significant difference in behaviour change between the intervention and control group more often targeted the response efficacy construct (7/7) than those which did not find significant differences (2/5).”

4. Your appendix currently lists data collection and exclusion criteria in a basic format. It would be more informative to include the specific standards directly in this figure.

We are unsure what the reviewer is referring to here as there is no appendix attached. If the reviewer would like to clarify, we would be very happy to respond.

We had meant to upload a supplementary file detailing the full search strategy with the original submission but it seems like this was missed. We have included this in the revised manuscript.

---

## [Decision Letter · Decision Letter 1]

28 May 2025

PONE-D-24-22494R1Using Protection Motivation Theory to support patient adherence in healthcare settings: A scoping reviewPLOS ONE

Dear Dr. Hawksworth,

Thank you for submitting your manuscript to PLOS ONE. After careful consideration, we feel that it has merit but does not fully meet PLOS ONE’s publication criteria as it currently stands. Therefore, we invite you to submit a revised version of the manuscript that addresses the points raised during the review process. There are additional reviewer queries that need attention. Please review and revise the manuscript accordingly.

We look forward to receiving your revised manuscript.

Kind regards,

Chenfeng Xiong

Academic Editor

PLOS ONE

Journal Requirements:

Reviewers' comments:

Reviewer's Responses to Questions

**Comments to the Author**

1. If the authors have adequately addressed your comments raised in a previous round of review and you feel that this manuscript is now acceptable for publication, you may indicate that here to bypass the “Comments to the Author” section, enter your conflict of interest statement in the “Confidential to Editor” section, and submit your "Accept" recommendation.

Reviewer #1: All comments have been addressed

Reviewer #3: (No Response)

2. Is the manuscript technically sound, and do the data support the conclusions?

Reviewer #1: Yes

Reviewer #3: Yes

3. Has the statistical analysis been performed appropriately and rigorously? 

Reviewer #1: Yes

Reviewer #3: Yes

4. Have the authors made all data underlying the findings in their manuscript fully available?

Reviewer #1: Yes

Reviewer #3: Yes

5. Is the manuscript presented in an intelligible fashion and written in standard English?

Reviewer #1: Yes

Reviewer #3: Yes

6. Review Comments to the Author

Reviewer #1: (No Response)

Reviewer #3: Summary:

The paper presents a scoping review on the use of Protection Motivation Theory (PMT) to support patient adherence in healthcare settings. The authors aim to map existing literature and offer practical recommendations for applying PMT-based strategies to improve health behavior outcomes. The topic is timely and highly relevant as patients' non-adherence presents significant challenges for healthcare systems.

Strengths:

1. The manuscript addresses an important gap by synthesizing literature on existing interventions on PMT.

2. The methods section is well-detailed and provides a clear description of the process.

Recommendations and Suggestions:

1. Although the authors provide a description of the main constructs of Protection Motivation Theory (PMT), I recommend including a conceptual figure that visually represents these components. A clear diagram of the PMT model would help understand the theory’s core constructs (e.g., threat appraisal, coping appraisal) and how they are defined and operationalized across the reviewed studies.

2. The manuscript would benefit from a clear explanation of why a scoping review was chosen over a systematic review or meta-analysis. A brief justification in the Methods or Introduction section would clarify the scope and intentions of the review for the reader.

3. One of the stated aims of the review is to provide ‘practical guidance for frontline healthcare professionals’ on applying PMT-based interventions. This aim should be more explicitly addressed in the Discussion section.

7. PLOS authors have the option to publish the peer review history of their article (what does this mean? ). If published, this will include your full peer review and any attached files.

**Do you want your identity to be public for this peer review?** For information about this choice, including consent withdrawal, please see our Privacy Policy .

Reviewer #1: No

Reviewer #3: No

---

## [Author Response · Author response to Decision Letter 2]

14 Jul 2025

Dear Editor,

Thank you for the opportunity to submit a revised draft of the manuscript “Using Protection Motivation Theory to support patient adherence in healthcare settings: A scoping review” for publication in PLOS ONE. We have incorporated the suggestions made by the reviewer and these are highlighted within the manuscript. Please see below a point-by-point response to the reviewers’ comments. All line numbers refer to the Revised Manuscript with Track Changes file.

1. Although the authors provide a description of the main constructs of Protection Motivation Theory (PMT), I recommend including a conceptual figure that visually represents these components. A clear diagram of the PMT model would help understand the theory’s core constructs (e.g., threat appraisal, coping appraisal) and how they are defined and operationalized across the reviewed studies.

We agree that this will be a helpful addition to the manuscript. We have now included Figure 1 which depicts the core constructs within PMT.

2. The manuscript would benefit from a clear explanation of why a scoping review was chosen over a systematic review or meta-analysis. A brief justification in the Methods or Introduction section would clarify the scope and intentions of the review for the reader.

Thank you for highlighting this. We have strengthened the justification for this choice in the final paragraph of the introduction:

“Scoping reviews can be used to map concepts and research activity across a broad area to clarify methods and identify gaps to shape future studies. Our aim is not to assess effectiveness as in a systematic review, but to systematically characterise how PMT is being applied to improve adherence in physical healthcare contexts,” (Lines 76-78)

3. One of the stated aims of the review is to provide ‘practical guidance for frontline healthcare professionals’ on applying PMT-based interventions. This aim should be more explicitly addressed in the Discussion section.

Thank you for this comment. We have made our reflections on this more explicit in the discussion:

“The synthesis provides a breakdown of the existing PMT based interventions in a range of clinical areas, allowing healthcare professionals to select components that are relevant to and practicable in their setting”. (Lines 250-252)

“The most relevant constructs within PMT are likely to differ across health conditions and treatment regimens. Therefore, exploratory research should be undertaken prior to intervention development to identify the constructs most pertinent to the specific clinical context.” (Lines 255-257)

Sincerely,

Liv Hawksworth

---

## [Decision Letter · Decision Letter 2]

7 Aug 2025

Using Protection Motivation Theory to support patient adherence in healthcare settings: A scoping review

PONE-D-24-22494R2

Dear Dr. Hawksworth,

We’re pleased to inform you that your manuscript has been judged scientifically suitable for publication and will be formally accepted for publication once it meets all outstanding technical requirements.

Kind regards,

Chenfeng Xiong

Academic Editor

PLOS ONE

Additional Editor Comments (optional):

Reviewers' comments:

Reviewer's Responses to Questions

**Comments to the Author**

1. If the authors have adequately addressed your comments raised in a previous round of review and you feel that this manuscript is now acceptable for publication, you may indicate that here to bypass the “Comments to the Author” section, enter your conflict of interest statement in the “Confidential to Editor” section, and submit your "Accept" recommendation.

Reviewer #3: All comments have been addressed

2. Is the manuscript technically sound, and do the data support the conclusions?

Reviewer #3: Yes

3. Has the statistical analysis been performed appropriately and rigorously? 

Reviewer #3: Yes

4. Have the authors made all data underlying the findings in their manuscript fully available?

Reviewer #3: Yes

5. Is the manuscript presented in an intelligible fashion and written in standard English?

Reviewer #3: Yes

6. Review Comments to the Author

Reviewer #3: (No Response)

7. PLOS authors have the option to publish the peer review history of their article (what does this mean? ). If published, this will include your full peer review and any attached files.

**Do you want your identity to be public for this peer review?** For information about this choice, including consent withdrawal, please see our Privacy Policy .

Reviewer #3: No

---

## [Editor Report · Acceptance letter]

PONE-D-24-22494R2

PLOS ONE

Dear Dr. Hawksworth,

I'm pleased to inform you that your manuscript has been deemed suitable for publication in PLOS ONE. Congratulations! Your manuscript is now being handed over to our production team.

Kind regards,

on behalf of

Dr. Chenfeng Xiong

Academic Editor

PLOS ONE